# Is It the Best Option? Robotic Surgery for Endometriosis

**DOI:** 10.3390/life14080982

**Published:** 2024-08-05

**Authors:** Jihyun Lee, Seongmin Kim

**Affiliations:** Gynecologic Cancer Center, CHA Ilsan Medical Center, CHA University College of Medicine, 1205 Jungang-ro, Ilsandong-gu, Goyang-si 10414, Gyeonggi-do, Republic of Korea; carolee0210@naver.com

**Keywords:** endometriosis, robotic-assisted laparoscopic surgery, conventional laparoscopic surgery

## Abstract

Endometriosis is a chronic condition affecting approximately 10% of women of reproductive age, leading to significant physical and emotional stress. Treatments include medical management and surgical interventions, with laparoscopic surgery being the gold standard for removing endometrial tissue. The advent of robotic-assisted laparoscopic surgery (RALS) has enabled more complex procedures to be performed minimally invasively, increasing its use in high-difficulty surgeries. Developed in the late 20th century, systems like the Da Vinci Surgical System have revolutionized surgery by enhancing precision, dexterity, and visualization. The latest models, including the Da Vinci Xi and SP, offer advanced features such as enhanced arm mobility, fluorescence imaging, and single-port capabilities. Comparative studies of RALS and conventional laparoscopy (LPS) for endometriosis show mixed results. While some studies indicate no significant differences in complications or recovery outcomes, others highlight longer operative times and hospital stays for RALS. Despite these drawbacks, RALS is not inferior to LPS overall. The clinical benefits of RALS include greater precision and accuracy, reduced surgeon fatigue, and a faster learning curve, facilitated by advanced ergonomic and control systems. However, the high costs and extensive infrastructure requirements limit the accessibility and availability of robotic surgery, particularly in smaller or rural hospitals. The absence of tactile feedback remains a challenge, though upcoming advancements aim to address this. Continued research and development are essential to make robotic surgery more cost-effective and broadly accessible, ensuring its benefits can reach a wider patient population. This abstract encapsulates the key aspects of robotic surgery’s development, comparative studies with conventional methods, and its clinical benefits and limitations, highlighting the need for ongoing improvements and research.

## 1. Introduction

Endometriosis is a chronic condition characterized by the presence of endometrial-like tissue outside the uterus [1]. It affects approximately 10% of women of reproductive age and can cause symptoms such as chronic pelvic pain, dysmenorrhea (painful periods), dyspareunia (pain during intercourse), and infertility [2]. The impact on quality of life is significant, leading to physical discomfort, emotional stress, and substantial healthcare costs [3]. Treatment for endometriosis typically includes medical management and surgical intervention. Medical management often involves hormonal therapies and pain management to alleviate symptoms [4]. Surgical management of endometriosis, particularly laparoscopic surgery, aims to remove endometrial implants and restore normal anatomy [5]. Laparoscopic surgery has been the gold standard for this purpose, allowing for minimally invasive access to the pelvic cavity [6]. Endometriosis surgery is primarily performed as minimally invasive surgery, which offers the advantages of shorter recovery times and reduced risks of pain and complications. Due to the anatomical structure of the major organs affected by endometriosis and the deeply infiltrative nature of the disease, endometriosis surgery is often highly complex and technically challenging. In particular, deep infiltrating endometriosis can spread to multiple organs such as the intestines, bladder, and ureters, making complete removal difficult. Therefore, some endometriosis surgeries fall under high-difficulty surgeries.

The use of RALS in endometriosis surgery is increasingly common. This approach enhances precision and accuracy in complex surgeries and reduces surgeon fatigue. Robotic systems offer enhanced depth perception, wrist articulation, and dexterity, particularly beneficial for complex cases or challenging anatomical locations, making them effective in treating deep infiltrating endometriosis [7]. The introduction of robotic surgery has made it easier to perform more complex procedures, not only for surgeries involving severe adhesions in benign conditions but also for oncological procedures [8].

## 2. Brief History of the Development of Robotic Surgery

Robotic-assisted laparoscopic surgery represents a significant advancement in surgical technology. The development of robotic surgery began in the late 20th century, initially driven by the needs of minimally invasive procedures. Key milestones include the introduction of the Da Vinci Surgical System, approved by the FDA in 2000, which revolutionized the field by providing surgeons with enhanced capabilities [9]. The Da Vinci Surgical System, developed by Intuitive Surgical, revolutionized surgery by providing surgeons with enhanced capabilities, including high-definition 3D vision, wristed instruments that bend and rotate in far greater ranges than the human hand, and an ergonomic console that allows surgeons to operate while seated. This initial system laid the groundwork for future innovations by providing surgeons with enhanced precision, dexterity, and control [10]. Over the years, advancements in robotic technology have expanded its applications, making it a valuable tool in various surgical disciplines, including gynecology. In 2006, the Da Vinci S system was introduced with improved visualization and more intuitive instrument controls, enhancing the surgical experience. This evolution continued with the 2009 launch of the Da Vinci Si system. In 2014, the fourth-generation Da Vinci Xi was introduced. This model offered enhanced arm mobility and flexibility, enabling access to multiple quadrants of the body without needing to reposition the patient [11]. This innovation significantly improved surgical efficiency and expanded the range of procedures that could be performed robotically. The most recent addition to the Da Vinci family came in 2018 with the introduction of the Da Vinci SP system. Designed specifically for single-port surgeries, this system improved cosmetic outcomes and reduced recovery times by allowing surgeons to perform complex procedures through a single small incision. This model integrates advanced imaging techniques, such as fluorescence imaging, to better visualize tissues and blood flow. The Da Vinci SP system combines the benefits of robotic precision with the advantages of minimally invasive surgery, further pushing the boundaries of what can be achieved in surgical care [12].

Building on the advancements of the Da Vinci systems, several novel robotic surgical systems have been developed in recent years. The Versius System was launched by CMR Surgical in 2019, highlighting its ergonomic controls and modular, portable design that can be adapted to various surgical settings. This system incorporates AI to provide feedback and learning opportunities, helping surgeons continuously improve their techniques [13]. In 2021, Medtronic introduced the Hugo RAS system, which features a modular design for flexible setup and customization. Its open console design provides the surgeon with an immersive 3D view while maintaining situational awareness in the operating room. This system aims to reduce the overall cost of robotic surgery systems, making advanced surgical techniques more accessible [14]. The same year, Johnson & Johnson developed the Ottava system, incorporating multiple independent arms mounted to the operating table, providing high-definition 3D visualization and advanced imaging techniques [15]. Additionally, Vicarious Surgical began the development of their system in 2021, utilizing virtual reality to create a highly immersive and intuitive surgical interface, with plans for FDA submission by 2026 [16]. It is designed to be compact and agile, enabling it to perform complex maneuvers within tight anatomical spaces. Many other systems are still under development. Robotic systems are becoming more modular, allowing for customizable setups tailored to specific surgeries or surgical environments.

Robotic-assisted surgery, with its rapidly evolving technology, has been successfully used in various types of surgeries, including gynecologic, urologic, cardiothoracic, and general surgeries. Clinical outcomes have generally shown benefits, such as reduced blood loss, shorter hospital stays, and quicker recovery times, though these advantages can vary depending on the type of surgery and patient population [17]. In the context of endometriosis, robotic surgery is increasingly used for complex cases, particularly those involving deep infiltrating endometriotic lesions that are difficult to treat with conventional laparoscopy.

## 3. Current Application and Outcomes on Endometriosis

### 3.1. Comparison of Surgical Outcomes

There are many studies comparing robotic surgery and laparoscopic surgery (Table 1). Most of these studies are retrospective, with only three being prospective. The first prospective study is the LAROSE trial [18]. The LAROSE (Laparoscopy vs. Robotic Surgery for Endometriosis) trial is one of the prospective studies comparing RALS and LPS. The study showed no significant differences in surgical time and complication rates between the two methods, although robotic surgery was associated with increased blood loss. Additionally, there were no significant differences in most aspects of postoperative quality of life. Considering that there were more patients with lower American Society of Reproductive Medicine (ASRM) scores in the robotic group, this could suggest that robotic surgery is inferior to laparoscopy. The second prospective study was reported in France in 2020, comparing the two groups in colorectal endometriosis [19]. There were no differences in characteristics between the two groups. Although robotic surgery had a longer operating time, no other differences were observed.

Most of the current evidence comes from retrospective studies, which provide valuable insights into the surgical outcomes of RALS and LPS. These studies collectively suggest that while RALS may have longer operative times and increased blood loss, it offers comparable or sometimes better outcomes in terms of recovery and complication rates. For instance, Nezhat et al. showed comparable outcomes for blood loss and hospital stay between RALS and LPS [20]. Chu et al. found no significant differences in most perioperative outcomes between the two methods [21]. There is another study suggesting that RALS might reduce recovery times in certain patient populations. These findings highlight the nuanced benefits and challenges of RALS in endometriosis surgery [22]. In addition to the mentioned studies, there are several other retrospective studies comparing the perioperative outcomes of RALS and LPS in the treatment of endometriosis.

### 3.2. Fertility Preservation or Residual Ovarian Function

A contemporary review highlights the role of robotic-assisted laparoscopy in reproductive surgeries. It discusses how robotic systems like the Da Vinci Surgical System provide enhanced dexterity and precision, which are critical for preserving fertility and maintaining ovarian function during complex reproductive procedures. The review emphasizes that robotic surgery is associated with better surgical outcomes in myomectomy and other fertility-preserving surgeries compared to traditional methods [34]. Recommendations for the surgical treatment of ovarian endometriomas emphasize the importance of preserving ovarian reserve during surgery. Robotic surgery, with its advanced precision and control, is suggested to potentially minimize damage to ovarian tissue compared to LSP. The use of robotic systems can be beneficial in complex surgeries where preserving ovarian function is critical [35].

A study compared robotic single-site (RSS) and single-port laparoscopic (SPL) surgeries in terms of fertility preservation. It found that the decrease in anti-Müllerian hormone (AMH) levels, an indicator of ovarian reserve, was significantly lower in the RSS group than in the SPL group. This suggests that robotic surgery might be more advantageous in preserving ovarian function, especially in complex cases such as endometriosis or larger, multilocular, or bilateral cysts with adhesion [36]. Another recent study investigated the impact of laparoscopic versus robotic cystectomy on ovarian tissue and follicular loss in patients with endometrioma, using AI-powered pathology analysis. The study, conducted on 28 patients, suggests that robotic cystectomy may result in less ovarian tissue and follicular loss compared to laparoscopic methods, particularly in cases of bilateral disease and larger cyst sizes. This research highlights the potential benefits of robotic surgery in preserving ovarian function during endometrioma cystectomy and recommends further studies to validate these findings [37]. These articles provide insights into how robotic surgery can play a beneficial role in preserving fertility and maintaining ovarian function, particularly in complex gynecological procedures.

### 3.3. Robotic Systems and Deep Infiltrating Endometriosis

Deep infiltrating endometriotic lesions often require meticulous dissection and precise excision, tasks that are difficult to perform with conventional laparoscopy due to limited instrument maneuverability and visualization. RALS provides enhanced 3D visualization, greater dexterity with articulated instruments, and improved ergonomics for the surgeon. These features make RALS particularly suited for addressing the complexities of deep infiltrating endometriosis, potentially leading to more complete lesion removal and better patient outcomes.

A meta-analysis comparing RALS to LPS for treating deep endometriosis analyzed fourteen studies with a total of 2709 patients [38]. The findings indicated no significant differences between RALS and LPS in terms of intraoperative and postoperative complications, conversion rates, or estimated blood loss. However, RALS was associated with longer operative times and longer hospital stays. Despite these drawbacks, robotic surgery turned out not inferior to laparoscopy in overall surgical outcomes. The study suggests the potential benefits of integrating new technologies with robotic platforms and recommends further prospective studies to validate these findings and improve scientific evidence. Another systematic review and meta-analysis evaluated the advantages of RALS compared to LPS in endometriosis surgery [39]. It concluded that RALS did not show significant advantages over LPS in terms of intraoperative and postoperative complications, estimated blood loss, or length of hospital stay. RALS was found to have longer operative times. The study highlighted the need for more comprehensive evaluations of long-term outcomes, such as pain relief, quality of life, and fertility results.

Additionally, there is an ongoing ROBEndo trial, scheduled to be completed by 2026, comparing robotic and conventional laparoscopy in deep infiltrating endometriosis, conducted by a Finnish group [40]. This prospective, randomized, controlled clinical trial will be conducted in a single-center setting. Patients with deep endometriosis verified by MRI and requiring surgery will be considered eligible for the study. A total of 70 patients will be allocated in a 1:1 ratio to receive either robotic-assisted or conventional laparoscopic surgery. The primary outcome will assess the surgical outcomes concerning pain symptoms. The result of this study is expected to contribute to a better comparison of robotic surgery to laparoscopic surgery in the field of deep infiltrative endometriosis.

However, these data are not satisfying for several reasons. Firstly, the severity of the disease varies in each study, making it difficult to be objective. Secondly, the data on conventional laparoscopy used in these studies mostly come from already experienced surgeons, making it hard for the relatively new robotic surgery to surpass that level of proficiency. Thirdly, surgical outcomes are not just about speed. It is difficult to prove the qualitative aspects of surgery through these studies.

## 4. Clinical Benefits of Robotic Surgery

### 4.1. Precision, Accuracy, and Ergonomics

RALS offers enhanced dexterity and range of motion, advanced imaging, navigation systems, and optoelectronic tracking technologies, allowing for more precise surgical maneuvers [41]. These advancements underscore the potential of robotic systems to enhance surgical outcomes, reduce recovery times, and increase the safety and effectiveness of complex procedures. Studies have shown that robotic-assisted surgeries, such as those performed with the Da Vinci and other systems, achieve high levels of accuracy [42]. For example, robotic-assisted spinal surgeries have demonstrated impressive precision in implant placement, with significantly reduced error margins compared to traditional methods [43]. In endometriosis surgery, where the accurate identification and excision of endometrial lesions are crucial, magnification capabilities and fine articulation enhance the ability to address deep infiltrating endometriotic lesions and reduce the risk of incomplete lesion removal and subsequent recurrence. Ergonomic surgery is also a huge benefit of robotic surgery. In LPS, when performing ureterolysis deep in the pelvis, the surrounding bowel can raise concerns about thermal damage, and on the left side, accessibility can be hindered by the adnexa. These are the drawbacks of using non-articulating instruments (Figure 1A,B). However, in RALS, the strength of the robotic arm and the utilization of articulation reduce the obstructions to the approach to the deep pelvis (Figure 2A,B). These could be benefits of robotic surgery.

### 4.2. Reduced Surgeon Fatigue

Robotic surgery allows surgeons to operate from a seated position at a console, reducing the physical burden compared to traditional laparoscopic surgery [44]. This ergonomic advantage is significant in long and complex endometriosis surgeries, where sustained concentration and reduced fatigue can improve surgical performance and patient outcomes. Studies have shown that surgeons using RALS experience lower stress levels and improved heart rate variability, with measurements indicating significantly lower physiological stress and reduced fatigue compared to standard laparoscopic surgery. This not only enhances the surgeon’s performance but also reduces the risk of errors due to fatigue compared to traditional surgical methods. Overall, these benefits are contributing to the growing adoption of robotic-assisted techniques in various surgical specialties.

### 4.3. Fast Learning Curve

The learning curve for robotic surgery is often shorter and more efficient compared to conventional laparoscopy, especially for complex procedures like endometriosis surgery. The intuitive control systems and enhanced dexterity provided by robotic platforms facilitate quicker skill acquisition and improved surgical outcomes [45]. Studies have shown that using a pelvic trainer and the Da Vinci Robotic Surgical System allows surgeons to acquire robotic-assisted laparoscopic skills more quickly and with greater precision than manual laparoscopic skills [46]. Furthermore, experience in laparoscopic surgery significantly improves performance in robotic surgery, indicating that surgeons can adapt to robotic systems more quickly if they already possess basic laparoscopic skills [47]. As a result, robotic surgery not only offers clinical advantages but also facilitates faster and more effective training for surgeons, ultimately improving surgical outcomes and expanding the capabilities of minimally invasive surgery.

### 4.4. Utilization of Single-Site Surgery

Single-site surgery’s limitations include restricted instrument movement, limited triangulation, and increased difficulty in suturing and complex dissection. These factors can make single-site surgery technically challenging and may limit its effectiveness in some complex cases. However, the Da Vinci SP system offers enhanced maneuverability resembling multi-port surgery and mitigates some of these challenges, making it a viable option for patients who prefer single-site surgery for cosmetic reasons, providing a feasible solution for complex endometriosis cases. However, data presented from Korea in 2018 [48] indicated that it was difficult to confirm any significant advantages of robotic single-site surgery for advanced-stage endometriosis. More studies are required to justify the non-inferiority of robotic-assisted surgical systems in the management of endometriosis.

## 5. Limitations and Challenges

### 5.1. Absence of Tactile Sense, Forced Feedback

One of the limitations of robotic surgery is the lack of tactile feedback, which is important for differentiating fibrotic tissues in endometriosis. An interesting study proposed a method using indocyanine green (ICG) in 15 areas to distinguish deep endometriosis lesions during robotic surgery. Injection of ICG intravenously allows for better identification of pelvic endometriosis, potentially overcoming the lack of tactile feedback [49]. Encouragingly, advancements such as the upcoming fifth-generation Da Vinci system aim to address the absence of haptic sensation with new force feedback features, enhancing the surgeon’s ability to distinguish and excise endometriotic lesions [50].

### 5.2. Accessibility and Availability

The initial acquisition cost of robotic surgical systems, such as the Da Vinci system, is very high [51]. These high costs and extensive infrastructure requirements limit the accessibility of robotic surgery, particularly in smaller or rural hospitals. As a result, robotic surgery systems are predominantly found in well-funded hospitals and urban centers [52]. This disparity in availability means that patients in rural or underfunded areas often do not have access to the benefits of robotic surgery, such as reduced recovery times and lower complication rates [53]. Additionally, the limited number of systems can lead to scheduling difficulties, further restricting access for patients who might benefit from these advanced surgical options [54]. Technological issues, such as system malfunctions or the need for constant software updates, can disrupt surgical schedules and reduce overall efficiency as well [55]. Efforts are ongoing to make these technologies more cost-effective and widely available, ensuring that the benefits of robotic surgery can reach a broader patient population, including those with endometriosis.

## 6. Conclusions

In conclusion, robotic-assisted laparoscopic surgery (RALS) for endometriosis offers significant advantages in precision, reduced surgeon fatigue, and a quicker learning curve, ultimately enhancing surgical outcomes and efficiency. Despite these benefits, the high costs and extensive infrastructure requirements of current robotic systems limit their accessibility, especially in smaller or rural hospitals. Additionally, the absence of tactile feedback in current systems presents challenges, although upcoming advancements such as the 5th generation Da Vinci system are expected to mitigate this issue. While robotic surgery has revolutionized the approach to complex surgical procedures, it is imperative to continue efforts to make these technologies more affordable and accessible. Future research and development are essential to overcoming existing limitations and ensuring that the benefits of robotic surgery are available to a broader patient population, including those in underfunded areas.

## Figures and Tables

**Figure 1 life-14-00982-f001:**
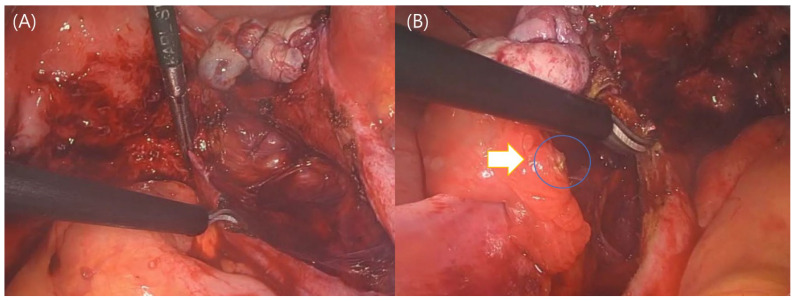
Drawbacks of non-articulating instruments in conventional laparoscopic surgery. (**A**) Scene of performing right pelvic peritonectomy. There is a concern about thermal damage to the rectum located right next to the monopolar. (**B**) Scene of performing left pelvic peritonectomy, with concerns about damage to the adjacent adnexa. The arrow shows actual thermal damage at the end of the fimbria.

**Figure 2 life-14-00982-f002:**
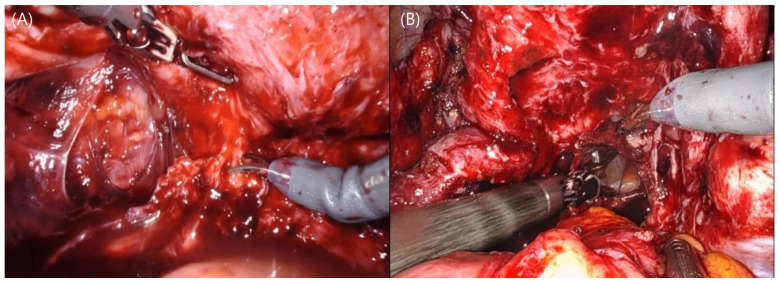
Ergonomic benefit of robotic surgery. (**A**) Utilizing the strength of the robotic instrument to lift the uterus, the obliterated posterior cul-de-sac can be easily dissected, and immediate hemostasis can be achieved using the monopolar. (**B**) Even the hard-to-reach deep pelvis can be easily accessed using the articulating instrument, with reduced concerns about thermal damage to surrounding organs.

**Table 1 life-14-00982-t001:** Summary of current evidence comparing RALS and LPS.

Study	Design	Group		Age		rASRM Stage	Operation Time (min)		Blood Loss (mL)		Hospital Stay (Days)	
		RALS	LPS	RALS	LPS		RALS	LPS	RALS	LPS	RALS	LPS
Nezhat et al. [20] (2010)	Retrospective	40	38	35	33	I–IV	191 (135–295)	159 (85–320)	60	65	N/A	N/A
Chu et al. [21] (2011)	Retrospective	25	96	N/A	N/A	III–IV	238 (120–630)	190 (71–674)	No difference	No difference	No difference	No difference
Dulemba et al. [22] (2013)	Retrospective	180	100	32.6	29.2	I–IV	77.4 ± 41.6	72.0 ± 28.5	2.92 ± 43.2	24.9 ± 24.3	N/A	N/A
Nezhat et al. [23] (2014)	Retrospective	32	86	42.5	40	III–IV	250 (176–328)	173 (123–237)	100	50	1	1
Nezhat et al. [24] (2015)	Retrospective	147	273	30	31	III–IV	196	135	40	25	>1	1
Magrina et al. [25] (2015)	Retrospective	331	162	40	38.3	III–IV	139 (40–531)	113 (28–347)	92	82	1.1	0.7
Soto et al. [18] (2017)	Prospective	35	38	34.3	34.5	I–IV	106 ± 48	101 ± 63	100 ± 229	43 ± 39	N/A	N/A
Le Gac et al. [19] (2020)	Prospective	23	25	36	37	III–IV	221 ± 94	163 ± 83	130 ± 86	108 ± 99	8.0 ± 4.4	6.5 ± 2.6
Raimondo et al. [26] (2021)	Retrospective	22	22	38	36	III–IV	207 ± 79	177 ± 63	181 ± 214	144 ± 101	8 ± 7	6 ± 2
Hiltunen et al. [27] (2021)	Retrospective	18	76	N/A	N/A	I–IV	N/A	N/A	N/A	N/A	N/A	N/A
Legendri et al. [28] (2022)	Retrospective	26	28	36.5	34	IV	N/A	N/A	N/A	N/A	N/A	N/A
Ferrier et al. [29] (2022)	Prospective	61	61	36	35	I–IV	208 ± 90	169 ± 81	No difference	No difference	N/A	N/A
Crestani et al. [30] (2023)	Retrospective	89	73	N/A	N/A	III–IV	N/A	N/A	N/A	N/A	N/A	N/A
Verrelli et al. [31] (2023)	Retrospective	71	104	37.3	38.4	III–IV	150	105	N/A	N/A	No difference	No difference
Volodarsky Perel et al. [32] (2023)	Retrospective	97	451	37.3	37.9	III–IV	N/A	N/A	N/A	N/A	N/A	N/A
Bandala et al. [33] (2024)	Retrospective	37	56	47.3	35.8	I–IV	170 ± 52	125 ± 43.6	50 ± 47.3	50 ± 46.8	1	2

Values are expressed as mean ± standard deviation or number (confidence interval). LPS; conventional laparoscopy, N/A; not available, RALS; robotic-assisted laparoscopic surgery, rASRM; revised American Society for Reproductive Medicine.

## Data Availability

No new data were created or analyzed in this study. Data sharing is not applicable to this article.

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
