# Peer review of "Is It the Best Option? Robotic Surgery for Endometriosis"

_life, 2024, doi:10.3390/life14080982_

Round 1

Reviewer 1 Report

Comments and Suggestions for Authors

The manuscript brought up a very rare but interesting topic. In this review, the authors summarized the current application and outcomes of robotic-assisted laparoscopic surgery on endometriosis and discussed its clinical benefits and drawbacks. However, the manuscript needs better organization and a clearer focus under each subtitle before it can be published. Here are some suggestions.

1.     In the introduction, the authors stated that robotic surgery is increasingly used in minimally invasive/high-difficulty surgeries, but did not clarify if robotic surgery has been used in endometriosis surgery. Does endometriosis surgery belong to minimally invasive/high-difficulty surgeries? How many proportions (if there is any data) of robotic-assisted surgery have been applied in endometriosis?

2.     In session 2, line 48, the authors introduced the benefits of Da Vinci Surgical system. But, what is exactly Da Vinci Surgical system? When/how this system was designed? The system was successfully used/assisted in what kinds of surgeries? A description of the system needs to be provided.

 Line 66, I assumed that this paragraph was aimed to present several novel systems that were developed following Da Vinci with advanced features. It is better to introduce these systems as a timeline instead of listing them. A better transition between the two paragraphs is also recommended.

Afterward, a third paragraph is necessary to give a brief description: With the above development, the current application of robotic systems in the clinic... In general, what surgeries are using these systems? How are the outcomes? And better connect to endometriosis.

3.     Session 3 has to be better written and organized.

Line 83, from here, the authors will focus on endometriosis. So, probably changed the title to “Current application/outcomes on endometriosis”.

Line 87, LAROSE needs to be defined. “Robotic surgery for endometriosis”?

In 3.1, the authors nicely summarized the surgical outcomes in a table, but only discussed prospective studies. The table included mostly retrospective studies, which were not mentioned. Those studies cannot be ignored. 3.1 is a major part of this review.

If robotic surgeries ameliorate the symptoms or the lesion recovery rates, compared to normal surgery?

Lines 100-123, the rationale has to be improved. Deep infiltrating endometriotic lesions are usually hard to completely remove with general laparoscopic surgery. So, in this context, the authors would like to discuss if robotic systems can advance the outcomes. This part may even give a separate subtitle.

In addition, the authors directly used the evaluation from several systematic reviews. Are these studies listed in the table? If so, please point them out and give more detailed information.

4.     Sessions 4 and 5 stated the benefits and limitations of robotic techniques. These are mostly general clinical descriptions other than endometriosis surgery. The authors need to well link these to endometriosis. This part might be reduced due to its less association with endometriosis, whereas increase session 3.

5.     This may be a bit out of the scope. Is there survey data to show if patients prefer or would like to use robotic surgery in endometriosis?

Comments on the Quality of English Language

Minor editing of English language required.

Reviewer 2 Report

Comments and Suggestions for Authors

It would have been interesting if the authors presented some data about personal experience, in this type of pathology, to personalize the comments regarding the preferred approach.

The article carries out a pertinent and objective analysis of the robotic-assisted approach versus the laparoscopic approach in pelvic endometriosis, starting from a not very systematized meta-analysis of the various articles published in the specialized literature.

The first 2 chapters of the article provide an overview of the pathology of deep pelvic endometriosis, emphasizing the value of the mini-invasive approach in this type of pathology, and the contribution of robotic surgery, with its latest performance improvement acquisitions. including associated AI programs.

The next chapter carries out a synthetic scan of the comparative studies from the last 15 years, honestly emphasizing the lack of evidence of clear advantages for the robotic approach, but instead an increase in the duration of the operation and hospitalization, as well as the costs of the intervention, although the results strictly related to the operative act are equivalent in the 2 techniques.

Chapter 4 is dedicated to the advantages of the robotic approach, which at first glance are not so obvious, but in a deep analysis they can be important: precision, accuracy, and ergonomics of maneuvers, reduced surgeon fatigue and last but not least fast learning curve, which can constitute points strengths for this approach compared to the laparoscopic approach.

Through all the aspects presented, the article achieves a balanced debate of the balance of the robotic versus laparoscopic approach in the pathology of pelvic endometriosis.

Reviewer 3 Report

Comments and Suggestions for Authors I appreciate the opportunity to review the manuscript entitled “Is it the best option? Robotic surgery for endometriosis” submitted in the journal Life in Medicine. The authors established the review about the advantages and disadvantages of the use of the robotic surgery in endometriosis treatment. The authors also mention the comparison of the robotic surgery with the other type of surgery in treatment of endometriosis as well as the different aspects of robotic surgery (such as limitations, leaning curve etc…)   Some specific comments:   1.      Please prepare the abbreviation list. 2.      Please mention the opinions and conclusions of the other reviews/meta-analysis which the topic is the usage of the robotic surgery in the endometriosis treatment.     My opinion is that manuscript meets the criteria to be published in journal Life after accepting the above mentioned comments.  

Round 2

Reviewer 1 Report

Comments and Suggestions for Authors

The authors improved a lot of the manuscript. I have no new comments.